# Pathway selection between click and acyl transfer reactions driven by aminoacyl phosphates

Debjyoti Bhattacharjee [1,2], Arti Sharma[1,2], Kun Dai[3], Thejus Pramod[2,3], Lenard Saile[2,3], Ralf Thomann[1,4] & Charalampos G. Pappas [1,2,3] ✉

Covalent transformations in biology follow defined temporal sequences that regulate processes such as acylation and phosphorylation, yet achieving comparable temporal control in synthetic systems remains challenging. Here, we report an abiotic aqueous reaction network in which aminoacyl phosphate esters bearing alkyne groups undergo a programmed sequence of covalent transformations governed by peptide-based nucleophiles. Phenolic nucleophiles promote rapid copper-catalyzed azide–alkyne cycloaddition (CuAAC), whereas cysteine-containing peptides transiently coordinate copper via their thiol groups, delaying CuAAC and favoring thioester formation. Kinetic analysis reveals that thiol–copper coordination controls early pathway selection, while self-assembly prolongs intermediate lifetimes and enables subsequent transformations. Combining both nucleophiles within a single peptide yields a three-step cascade comprising thioester formation, diester generation, and CuAAC. Variation of the azide structure further tunes product selectivity beyond acyl transfer. Together, these results demonstrate how the interplay of reactivity and supramolecular organization can encode intrinsic temporal order into chemically driven reaction networks.

Living systems rely on both chemoselectivity and reaction sequence to control molecular transformations[1]. From cell division[2] to protein biosynthesis[3] and metabolic regulation[4], biological functions are often encoded in temporally ordered reaction sequences. In such networks, specific covalent transformations are triggered, delayed, or suppressed depending on the molecular context, ensuring that complex pathways unfold with the necessary spatial and temporal precision. Among various processes, protein biosynthesis[3] and its subsequent modifications exemplify how such sequential chemical transformations are controlled in biological systems. This process begins with activation of amino acids as aminoacyl-adenylates enabled by adenosine triphosphate (ATP), followed by their transfer to tRNAs and peptide bond formation on the ribosome[5]. Once the polypeptide chain is

assembled, proteins undergo additional covalent transformations, such as acetylation[6], lipidation[7], ubiquitination[8], or site-specific phosphorylation[9] of amino acid side chains (e.g., serine, threonine, or tyrosine), which regulate their folding and catalytic activity. Such phosphorylation[10] events are typically catalyzed by kinases that selectively recognize the chemical environment and side chain functionalities of target residues. These modifications frequently depend on prior states, with one transformation acting as a prerequisite or barrier for the next. Through this control, biology regulates structure and reactivity, ensuring that chemical transformations proceed in a defined order. Capturing this selective and temporal reaction sequence in abiotic networks[11–18] remains challenging. Despite progress in dynamic combinatorial chemistry[19–26], chemically driven

[1]Freiburg Center for Interactive Materials and Bioinspired Technologies (FIT), University of Freiburg, Freiburg, Germany. [2]Institute of Organic Chemistry, University of Freiburg, Freiburg, Germany. [3]DFG Cluster of Excellence livMatS @FIT—Freiburg Center for Interactive Materials and Bioinspired Technologies, University of Freiburg, Freiburg, Germany. [4]Freiburg Materials Research Center (FMF), University of Freiburg, Freiburg, Germany. ✉e-mail: charalampos.pappas@livmats.uni-freiburg.de

assembly[27–37] and supramolecular polymers[38–41], most synthetic systems rely on single-step transformations or require external stimuli, such as enzymes[42–45], pH shifts[46–49], reduction[50], sound[51], or light[52] to control different pathways and achieve specificity. Sequential transformations are particularly difficult to control when multiple reactive species coexist under identical conditions, often leading to competitive reactions, undesired side products and poor selectivity. Among various synthetic transformations, the copper-catalyzed azide–alkyne cycloaddition (CuAAC) widely known as "click chemistry" offers an exceptionally robust method for selective bond formation[53–56]. The development of strain-promoted variants[57] has further extended this chemistry into living systems, circumventing the need for cytotoxic copper catalysis. Today, click reactions are central to bioconjugation[58,59], polymer synthesis[60–62], and drug-discovery[63,64]. Moreover, click reactions are further utilized in lipid conjugation[65,66], for creating mechanically interlocked molecules[67], and in controlling photo-switching behavior[68]. Despite their broad applicability, orthogonal reactions often remain sensitive to their surrounding chemical environment, which can impact both efficiency and selectivity[69]. Despite its lack of inherent dynamicity relative to biological reaction networks, CuAAC offers a uniquely robust and orthogonal transformation that enables us to study the influence of supramolecular organization on chemical reactivity. Variations in local microenvironment can regulate site accessibility and thereby govern the timing and ordering of covalent processes. Accordingly, we aim to establish systems in which reaction sequences are dictated by intrinsic molecular architecture and environmental cues rather than by external control.

Herein, we introduce an abiotic system in which two covalent transformations - acyl transfer and CuAAC take place simultaneously. Aminoacyl phosphate esters bearing alkynes serve as bifunctional reactants: they undergo acyl transfer with nucleophilic substrates such as tyrosine or cysteine-containing amino acids and peptides, while also engaging in click reactions with aliphatic or aromatic azides. Through variation of nucleophilic components and peptide sequences, we found that the temporal sequence of covalent bond formation is primarily governed by the chemical nature of the nucleophile. Tyrosine-based nucleophiles, bearing phenolic side chains, do not initially participate in acylation, as the phosphate ester instead undergoes rapid CuAAC with the azide. Ester products form only after the click reaction is complete (Pathway I, Fig. 1). An exception occurs with more hydrophobic tyrosine-containing peptides, which partially alter this sequence through self-assembly. However, even in such cases, the concentration of acylated species remains limited, and the temporal separation between pathways is narrow. In contrast, cysteine-containing peptides exhibit markedly different behavior. Their thiol side chains engage readily in thioester formation, which proceeds in high yield and transiently delays the onset of the click reaction (Pathway II, Fig. 1). This shift likely arises from both the higher nucleophilicity of thiols and their ability to transiently coordinate copper ions, reducing the availability of catalytically active Cu(I) and thereby deferring CuAAC. The delay is further amplified in peptide sequences enriched with an aromatic residue, where gelation imposes additional kinetic constraints. These sequential transitions can be tracked through macroscopic changes such as sol-gel-sol transitions, color shifts and spectroscopic signatures. Fmoc fluorescence is quenched upon triazole formation, offering a direct readout of CuAAC progression. Notably, the same principles extend beyond a single nucleophile or azide. We demonstrate that substrates bearing multiple nucleophilic groups, such as dipeptides featuring cysteine and tyrosine residues involved in multiple acyl transfer reactions prior to click chemistry. Moreover, when multiple azide inputs are introduced simultaneously, the system maintains reaction sequence and favors the selective formation of intermediates incorporating aliphatic azides. These findings reveal a further layer of regulation in which both chemical structures of the nucleophile and azide influence the reaction

sequence and guide the activation of a preferred pathway. Together, integrating acyl transfer and click chemistry within a single system shows how intrinsic molecular features can direct reaction sequences. Non-equilibrium intermediates, such as thioesters or esters, can then undergo post-acyl-transfer modifications, leading to pathway-dependent covalent outcomes.

## Results

### Click or Acyl transfer?

We previously demonstrated that aminoacyl phosphate esters enabled acyl transfer reactions to various nucleophilic substrates, giving rise to distinct reactivity patterns depending on side chain[70] and phosphate substitution[71]. In those systems, structural elements, such as the amino acid side chains or the nature of the phosphate ester dictated the activation and deactivation of pathways[72] selectively controlling the formation and lifetime of various intermediates. However, these systems typically allowed a single transformation, as the resulting products lacked additional functional groups for further covalent transformation. In contrast, the current system integrates a second reactive site, an alkyne moiety, enabling a CuAAC (click) reaction to compete or follow the acyl transfer. To minimize side reactions and avoid oligomerization[73], the aminoacyl phosphate ester and the nucleophilic partners (amino acids and dipeptides) were N-terminally capped. This ensured that covalent transformations arose exclusively from side-chain functionalities such as phenols or thiols. The alkyne-containing aminoacyl phosphate ester was synthesized from Fmoc-propargylglycine (Fmoc-Pra-OH) abbreviated as **Fmoc-Alkyne (1)**, where the Fmoc group served a dual purpose: providing an aromatic chromophore for spectroscopic monitoring and promoting aromatic stacking interactions which can influence both reaction kinetics and self-assembly. To investigate how sequence influences reactivity and phase behavior, we introduced charged (e.g., aspartic acid) or aromatic (e.g., phenylalanine) residues adjacent to the nucleophilic sites. All reactions were carried out in aqueous buffer, using $CuSO_4 \cdot 5H_2O$ and sodium ascorbate to generate Cu(I) in-situ, avoiding the instability of preformed Cu(I) species in water. For CuAAC reactions, we employed initially 6-azidohexanoic acid abbreviated as **azide**. Single letter codes for amino acids have been used throughout this work. The complete structures of the building blocks and their abbreviations are provided in the supplementary information. The reactions for tyrosine derivatives have been conducted at pH 10.0 whereas for the cysteine derivatives, pH 8.0 has been used, considering the pKa of the respective nucleophilic peptide[72].

We started with tyrosine-containing systems to examine how phenolic side chains influence the sequence of covalent transformations. A reaction mixture containing **Fmoc-Alkyne-EP (1-EP)** (10 mM), **azide** (20 mM) and **Ac-Y** (40 mM) in 0.2 M bicarbonate buffer (pH 10.0) yielded no detectable **Ac-Y-Ester (3a)** intermediate at the early stages (Fig. 2a, Supplementary Fig. 1). Instead, rapid occurrence of CuAAC were observed, evidenced by ultra-performance liquid chromatography coupled with mass spectrometry (UPLC-MS). These observations indicate a sequential transformation in which the aminoacyl phosphate ester is first modified via a triazole moiety to yield **Click-EP (2-EP)**, followed by slower ester formation with the phenolic nucleophile, resulting in **Ac-Y-Ester-click (4a)**. Finally, hydrolysis led to the formation of **Click (2)**. To assess whether the sequence of the reaction could be modulated by the amino acid residues, we next examined **Ac-YD** (Fig. 2b, Supplementary Fig. 2), a dipeptide containing tyrosine and aspartic acid.

Taking into account the negatively charged side chain of aspartic acid, we hypothesized that electrostatic repulsion with the phosphate moiety could influence both intrinsic reactivity and supramolecular assembly.[71] Supporting this hypothesis, only a small amount of **Ac-YD-ester (3b)** (-0.1 mM) accumulated within the first 30 min before being rapidly converted into the **Ac-YD-Ester-click (4b)** product. This

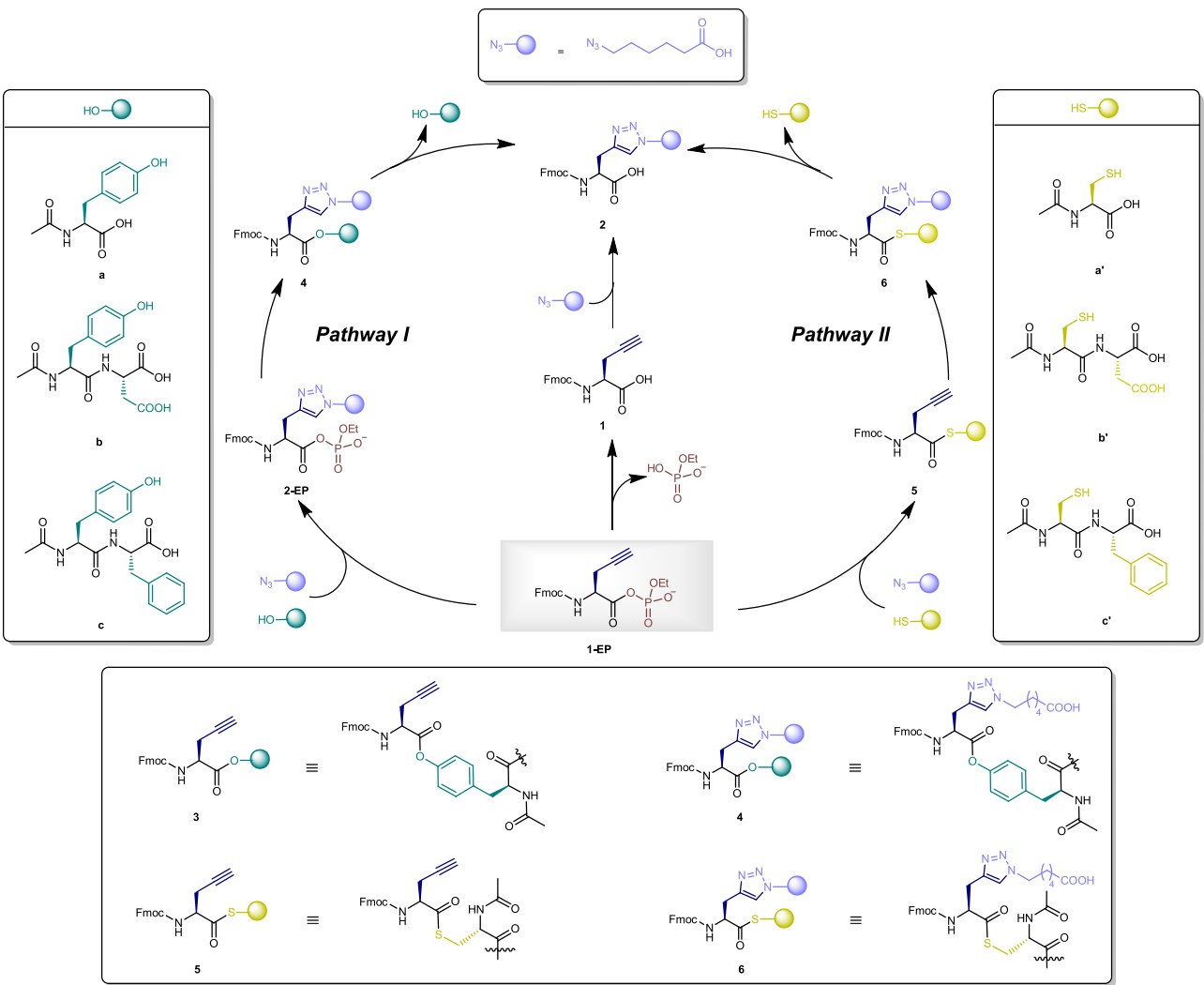

**Fig. 1 | Pathway selection between CuAAC and acyl transfer reactions.** Schematic representation of a chemical reaction network initiated by **Fmoc-Alkyne-EP (1-EP)** highlighting two different pathways based on the nucleophilic amino acidic residue used, exhibiting a temporal control between acyl transfer and CuAAC reaction.

behavior indicates that, in this system, acyl transfer proceeds more slowly than CuAAC. It further suggests that in the corresponding tyrosine system (**Ac-Y**), even if the ester intermediate **3a** forms transiently, it is likely consumed immediately to generate the click product **4a**, preventing its accumulation. Under these conditions, **1-EP** is first converted via CuAAC to ~5 mM of **2-EP**, followed by esterification to give **4b**, which ultimately undergoes hydrolysis to yield **2**. Together, these observations show that although acyl transfer and CuAAC can occur concurrently in one pot, CuAAC proceeds more rapidly when tyrosine is the nucleophile, owing to the relatively weak nucleophilicity of the phenolic side chain. Consequently, the faster kinetics of CuAAC limit the build-up of early acyl-transfer intermediates, which may also escape detection due to rapid conversion or hydrolysis on experimentally accessible timescales. As a consequence, multiple click-derived species, such as **2-EP, 4b**, and **2** coexist, reflecting the absence of a dominant intermediate and highlighting the lack of temporal separation between the two transformations. We then investigated whether self-assembly could modulate the sequence of covalent transformations. Using **Ac-YF** (Supplementary Figs. 3, 4), a more hydrophobic tyrosine-containing dipeptide, we observed rapid gelation within 5 min of initiating the reaction. After 30 min, product analysis revealed a more complex distribution: in contrast to **Ac-Y** or **Ac-YD**, which showed minimal ester formation, **Ac-YF** produced the acyl transfer product i.e., **Ac-YF-Ester (3c)** alongside CuAAC-derived

species. These findings suggest that while both pathways remain operative under these conditions, their relative kinetics are altered by the self-assembled environment, deviating from reaction **Pathway I** showed in Fig. 1. In case of **Ac-YF**, the aromatic side chain likely promoted co-assembly with the alkyne-functionalized phosphate ester via π–π interactions leading to early acylation. The resulting co-assemblies concentrate nucleophiles and electrophiles within the assembled phase, enhancing acyl transfer while simultaneously limiting the diffusion of azides into the reactive domain. Furthermore, because the CuAAC catalyst remains in the soluble phase, only unassembled fractions of **1-EP** can undergo click reaction. When the majority of reactive species are trapped within the assemblies, the exchange kinetics between assembled and soluble states become rate-limiting for CuAAC. This kinetic mismatch allows acyl transfer to proceed first, even though CuAAC remains the faster reaction in solution, effectively reversing the reaction sequence observed in unassembled systems. Nevertheless, multiple species, including **3c, 2** and **Ac-YF-Ester-click (4c)** coexist over an extended period of 7 days, indicating that self-assembly enables partial but not complete temporal separation between acylation and click pathways. Notably, even under assembling conditions, the yield of **3c** does not exceed ~2 mM, highlighting an intrinsic limitation of phenolic nucleophiles in promoting acyl transfer. This constraint highlights the need for alternative peptide sequences to facilitate broader windows of separation

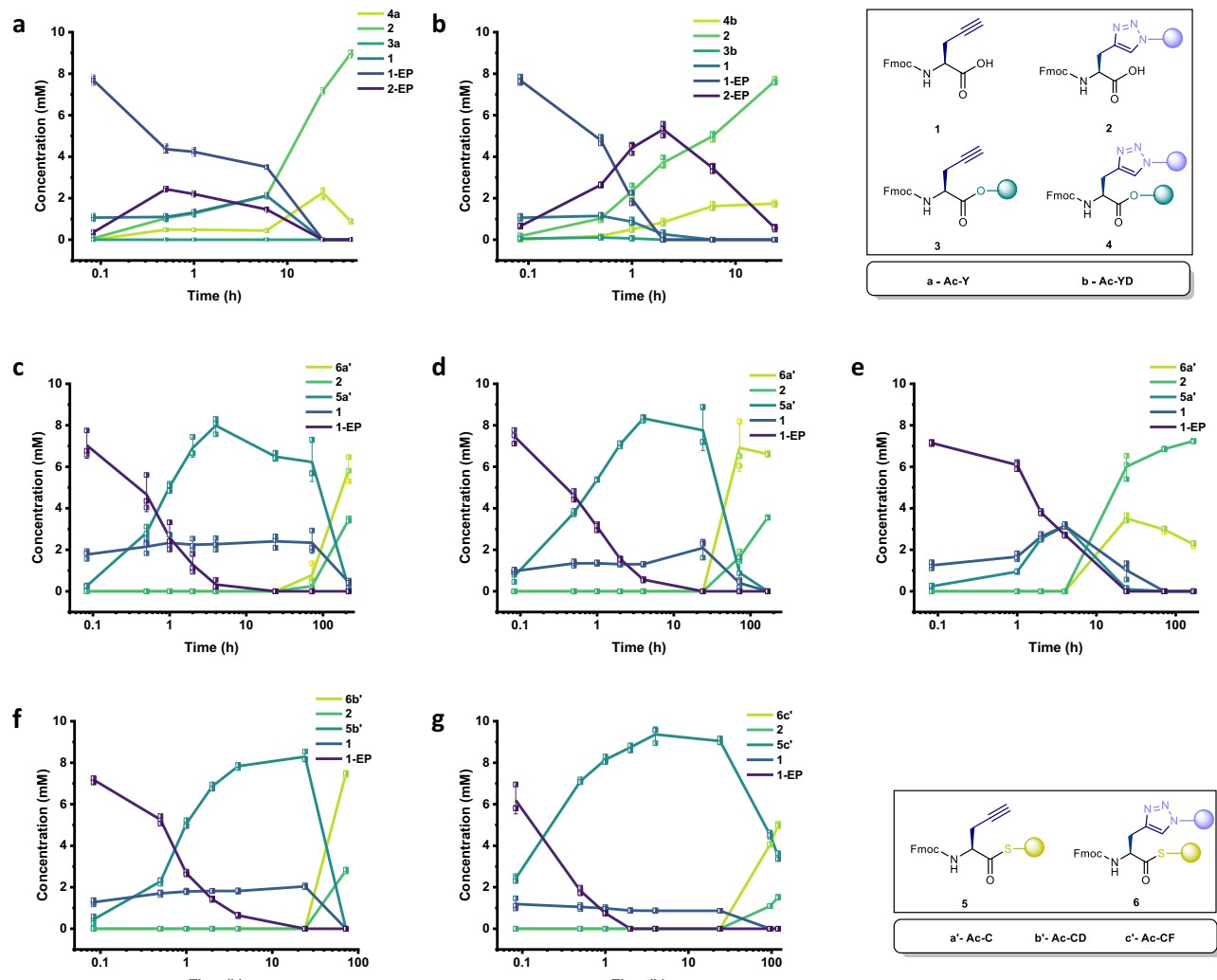

**Fig. 2 | Side-chain control of competing covalent pathways.** Time-course concentration profiles of tyrosine-containing substrates: **Ac-Y** (**a**) and **Ac-YD** (**b**), both at 40 mM with 10 mM **1-EP** and 20 mM **azide**. Time-course concentration profiles of cysteine-containing substrates: **Ac-C** (40 mM) reacted with varying azide concentrations, 10 mM (**c**), 20 mM (**d**) and 40 mM (**e**). In (**e**), the **Ac-C** concentration is also reduced to 20 mM. Time-course concentration profiles of cysteine dipeptides **Ac-CD** (**f**) and **Ac-CF** (**g**), each at 40 mM with 20 mM azide, highlighting the influence of amino acid side chains on reaction kinetics. In all graphs, error bars represent the standard deviation from three independent experiments.

between covalent transformations. Having characterized the behavior of tyrosine-based nucleophiles, we next investigated whether the observed temporal sequence between acyl transfer and CuAAC could be reprogrammed by altering the nucleophile. We turned to cysteine-containing derivatives, which feature thiol side chains with enhanced nucleophilicity[72] and the capacity for metal coordination[74]. Thiols readily interact with copper(II), forming [Cu(I)-SR] complexes that under aerobic conditions may regenerate Cu(II) with concurrent disulfide formation[75]. We hypothesized that transient copper sequestration could delay the onset of the CuAAC reaction, thereby allowing the acyl transfer to proceed first regardless of the presence or absence of a self-assembling environment. To examine this, we carried out a reaction containing **1-EP** (10 mM), **azide** (10 mM), and **Ac-C** (40 mM) in 0.2 M phosphate buffer at pH 8.0 (Fig. 2c). The buffer conditions were selected to minimize hydrolysis of the phosphate ester while preserving the high nucleophilicity of thiols. Moreover, previous studies have reported that CuAAC proceeds efficiently in aqueous media at pH 7-9[76]. A lower azide concentration was used to reduce the kinetic advantage of CuAAC, allowing potential acyl transfer intermediates to accumulate. During the first 72 h, we observed exclusive formation of the **Ac-C-**

Thioester (**5a′**) (~6.5 mM) intermediate, with no detectable CuAAC products. Beyond this point, the CuAAC reaction gradually initiated, yielding ~0.8 mM **Ac-C-Thio-click** (**6a′**) and ~ 0.3 mM **2**. Subsequently, the remaining early acylation intermediates converted into the corresponding click products (**6a′** and **2**). The delayed initiation of click reaction coupled with thioester formation indicates that cysteine enables temporal reordering of covalent transformations, consistent with the hypothesis that metal sequestration leads to delayed release of Cu(II). Notably, this transformation also coincided with weak gelation. As time progressed and CuAAC gradually initiated, the thioester-derived weak gel transitioned into a solution. This phase change revealed an additional layer of regulation: either the hydrogel restricted the diffusion of azide and copper catalysts, or the hydrophilic triazole products of the CuAAC reaction disrupted the assembled network. Given that hydrolyzed **1** was freely present from early time points without undergoing CuAAC, the latter scenario appears less likely. Moreover, our findings from tyrosine derivatives already indicate that self-assembly limits access to reactive sites and thereby modulating the onset of click reaction. To examine whether the delay in click reaction was influenced by **azide** concentration, we repeated the experiment using the same buffer and nucleophile (**Ac-**

**C** at 40 mM) but increased the azide concentration to 20 mM (Fig. 2d, Supplementary Figs. 5, 6). Under these conditions, **5a'** occurred at early time points, while the subsequent formation of **6a'** proceeded more efficiently, reaching ~7 mM conversion within 72 h. Additionally, we altered the stoichiometry by decreasing the nucleophile concentration to 20 mM while increasing azide to 40 mM (Fig. 2e). This alteration of stoichiometry did not change the sequence of events. However, CuAAC reaction began to occur after 24 h. These results show that while stoichiometry can modulate the kinetics and affect the delay between acyl transfer and CuAAC, it does not determine the overall sequence of these transformations. Next, we investigated how the system would behave at a higher pH, mimicking the conditions for the tyrosine systems. The reaction was kept using **1-EP** (10 mM), **azide** (10 mM), and **Ac-C** (40 mM) in 0.2 M bicarbonate buffer at pH 10.0 (Supplementary Fig. 6c) and a similar trend was observed as in pH 8.0. After establishing temporal control in Ac-C-based systems, we next examined how introducing different amino acids at the C-terminus of cysteine-containing dipeptides (**Ac-CX**) could influence the kinetics and the yield of the different species formed. Therefore, we investigated two cysteine-containing dipeptides, **Ac-CD** (Fig. 2f, Supplementary Figs. 7, 8) and **Ac-CF** (Fig. 2g, Supplementary Fig. 9). The aspartic acid residue in **Ac-CD** introduces electrostatic repulsion, while the phenylalanine in **Ac-CF** promotes assembly through aromatic stacking interactions, similar to the behavior observed in tyrosine-based systems. All reactions were performed under identical conditions, i.e., **1-EP** (10 mM), **azide** (20 mM), and nucleophile (**a´** / **b´** / **c´**) (40 mM) in 0.2 M phosphate buffer at pH 8.0, enabling direct comparison of the resulting concentration profiles over time. In case of **Ac-CD**, the transformation from acyl transfer to CuAAC proceeded efficiently, with full conversion to **Ac-CD-Thio-click** (**6b'**) within 72 h. No gelation was observed during the process, allowing free diffusion of azide and enabling the CuAAC reaction to proceed shortly after acyl transfer. Compared to **Ac-C**, the reaction advanced slightly faster. In contrast, the **Ac-CF** system showed a different behavior. Gelation occurred within 30 min of initiating the reaction, indicating the formation of assemblies. Although **Ac-CF-Thioester** (**5c'**) formed readily, indicating rapid acyl transfer, the onset of CuAAC was delayed relative to both **Ac-C** and **Ac-CD**. Even after 120 h, the reaction had not reached full conversion, pointing towards strong kinetic gating imposed by the assembled state (Fig. 2g). To evaluate the impact of concentration, we repeated the experiments using a lower peptide (20 mM) and a higher azide concentration (40 mM) (Supplementary Figs. 8, 10). Under these conditions, both the **Ac-CD** and **Ac-CF** systems showed an earlier onset of CuAAC following acyl transfer, similar to **Ac-C** (Fig. 2e). However, the temporal profiles of the systems remained distinct. If copper sequestration were the sole factor governing temporal control, one would expect identical delays across all cysteine-containing substrates. The persistent variation in timing among **Ac-C, Ac-CD** and **Ac-CF** instead suggests that self-assembly further modulates the temporal profile, amplifying the separation between thioester formation and CuAAC, by influencing the accessibility and diffusion of reactants. Additional support for this conclusion came from co-solvent experiments. We hypothesized that by diminishing hydrophobic interactions in the presence of organic solvent, the delay between acyl transfer and CuAAC could be affected. When reactions were conducted in a 1:1 mixture of DMSO and phosphate buffer (10 mM **1-EP**, 20 mM **azide**, 40 mM **Ac-C** / **Ac-CF**; Supplementary Figs. 11,12), the delay between thioester formation and click reaction was significantly reduced. To further probe the role of aggregation, we repeated the experiments at lower reactant concentrations (5 mM **1-EP**, 10 mM **azide**, 20 mM **Ac-C**; Supplementary Fig. 13), under which no gelation was observed. Interestingly, the same reaction sequence was maintained, with acyl transfer still preceding CuAAC. More broadly, across different systems, concentration-dependent

experiments and macroscopic observations indicate that gelation emerges only above a threshold acyl phosphate ester concentration and coincides with the accumulation of acyl-transfer intermediates (Supplementary Figs. 14–17), whereas lower concentrations favored homogeneous solutions. The hypothesis that thiol-mediated metal sequestration delays the onset of CuAAC, was further confirmed from experiments with excess copper sulfate (Supplementary Fig. 18a). Under these conditions, CuAAC proceeded exclusively and no acyl transfer was observed, supporting the idea that free copper overrides thiol coordination. We also examined the tyrosine-based system in the presence of an inert thiol, N-Boc-methionine (Supplementary Fig. 18b), which lacks nucleophilic reactivity yet can coordinate copper. Notably, the reaction profile remained unchanged compared to the system without the inert thiol (Fig. 2a), suggesting that copper coordination alone is insufficient to alter the reaction sequence without a reactive nucleophile. Overall, the two key factors that govern the observed reaction sequence are: (1) thiol-copper coordination delays catalyst availability, suppressing early CuAAC; and (2) self-assembly, particularly in peptides with aromatic side chains like **Ac-CF**, creates a phase mismatch between catalyst and substrate and slows azide diffusion, prolonging the lifetime of thioester intermediates. Together, these effects allow intrinsic molecular features and supramolecular organization to direct the order of covalent transformations.

## Spectroscopic and microscopic tracking of click and acyl transfer reaction progress

After identifying how different nucleophiles affect the timing and sequence of acyl transfer and CuAAC, we used spectroscopic and microscopic techniques to further support our kinetic findings. Libraries containing **Ac-C** and **Ac-CF** (40 mM), which formed gels, were initially colorless. Upon onset of the click reaction, these gels gradually turned yellow and eventually dissolved into solutions (Supplementary Fig. 19). For both **Ac-C** and **Ac-CF**, this color change was noticed after ~72 h, coinciding with the delayed appearance of CuAAC products.

In contrast, **Ac-CD**, which did not form a gel under the same conditions, exhibited an earlier color change, consistent with its faster transition from thioester (**5b´**) to the click product (**6b´**). Samples containing tyrosine species, where CuAAC dominates at the early stages, appeared yellow immediately upon mixing, further supporting that the visible color change correlates with the initiation of CuAAC reaction. To complement chromatographic analysis and monitor changes in the arrangement of the aromatics, we employed fluorescence spectroscopy. The Fmoc group, present on the aminoacyl phosphate ester (**1-EP**), provided an intrinsic fluorophore whose emission is highly sensitive to the aggregation state. In the absence of a nucleophile, addition of azide, copper sulfate and sodium ascorbate rapidly triggered CuAAC with **1-EP**, caused reduction in fluorescence intensity (Supplementary Fig. 20a). The emission spectrum of **1-EP** showed a peak around 320 nm, characteristic to the monomeric state of the acyl phosphate precursor. Upon CuAAC reaction, a reduction in the fluorescence intensity was observed within 15 min (Supplementary Fig. 20a), which is consistent with the complete conversion of **1-EP** into **2-EP** observed using UPLC. When a nucleophile such as **Ac-C** was present, a further decrease in fluorescence was observed during the early stages of the reaction (Supplementary Fig. 20b). This intensity fall corresponded to the formation of the thioester (**5a´**), suggests a differential arrangement of the fluorophores upon acyl transfer. Similarly, **Ac-CD** and **Ac-CF** (Supplementary Fig. 20c-d) were found to exhibit analogous trends. To gain insights into the supramolecular structures underlying the distinct reaction sequence, we employed transmission electron microscopy (TEM) (Fig. 3). These experiments provided direct visualization of the nanostructures formed by different nucleophile-peptide combinations at various time points in the presence of the aminoacyl phosphate ester. The phosphate ester itself showed

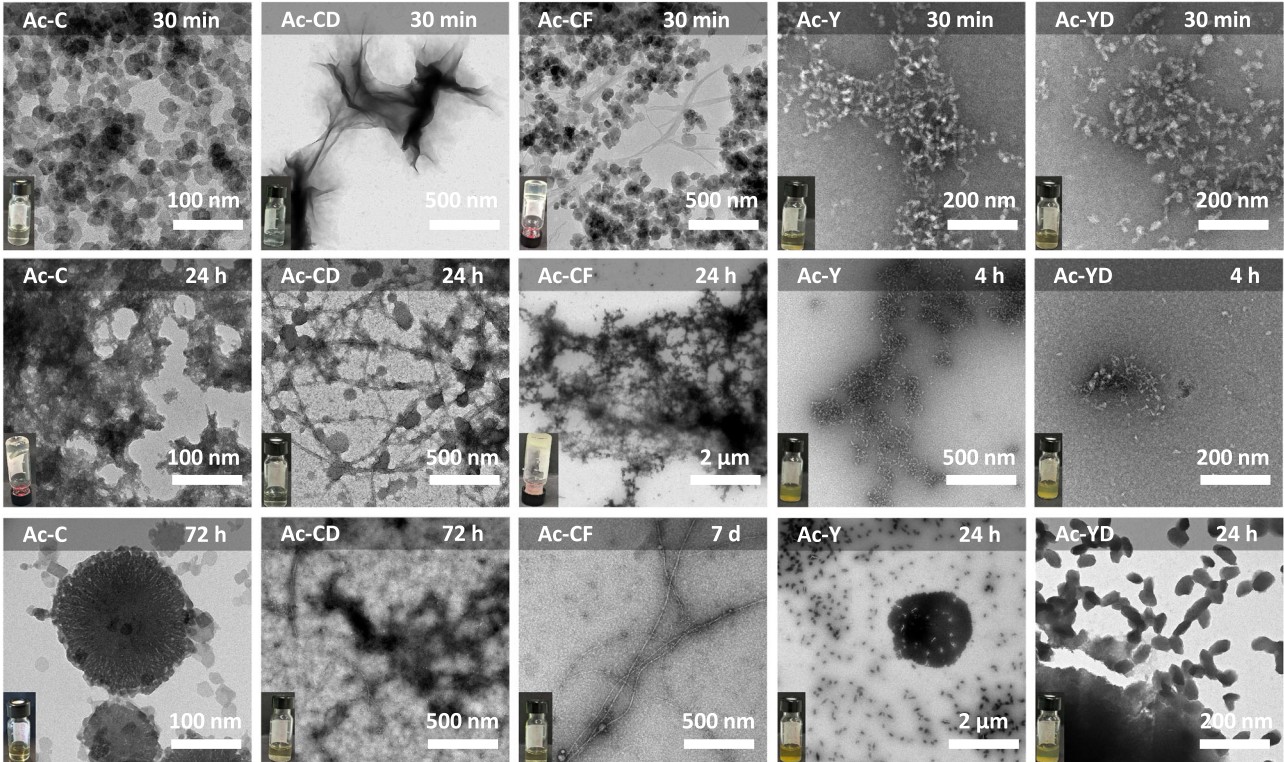

**Fig. 3 | Supramolecular assemblies from cysteine and tyrosine-based peptide substrates.** Transmission electron microscopy (TEM) images of assemblies formed from 10 mM **1-EP**, 20 mM **azide** and 40 mM peptidic nucleophiles in 0.2 M phosphate buffer (pH 8.0) and 0.2 M bicarbonate buffer (pH 10.0) for cysteine and tyrosine-containing residues, respectively, at different time points. Vial images represent macroscopic appearance of reactions at specific time points.

aggregation (Supplementary Fig. 21), consistent with its amphiphilic structure. Distinct morphologies appeared depending on the sequence and the nucleophile used from amino acid and peptide substrates (Supplementary Figs. 22–29). **Ac-C** and **Ac-CF**, which exhibited the strongest temporal delays in CuAAC, initially formed dense spherical aggregates (30 min), corresponding to the early formation of thioesters (**5a´** and **5c´**). Over-time, as the concentration of the thioester intermediate decreased, these aggregates underwent structural rearrangement: **Ac-C** gave rise to larger assemblies by 72 h, while **Ac-CF** assembled into fibers in the course of a week. In contrast, **Ac-CD** showed ill-defined aggregates at the end of the process (after click product i.e., **6b´** was formed), although spherical aggregates and small fibrillar structures were visualized at earlier stages of the process. Varying stoichiometry for **Ac-C** and **azide** has also resulted in different assemblies (Supplementary Figs. 25, 27). Tyrosine-based derivatives displayed different behavior. Both **Ac-Y** and **Ac-YD** showed no evidence of organized supramolecular assemblies at any time point, consistent with their rapid click reaction and minimal accumulation of ester intermediates. In these cases, the click reaction outpaces acyl transfer, leading to early formation of the triazole-modified product (**2-EP**), which lacks the structural features necessary to promote assembly. As a result, the final aggregates are amorphous, regardless of peptide sequence. The only exception among the tyrosine-containing systems was **Ac-YF**, which formed long, intertwined fibrils at early stages of the reaction. This behavior correlates with its enhanced ability to stabilize ester intermediates and induce gelation, likely driven by π–π stacking interactions contributed by the phenylalanine side chain. Confocal fluorescence microscopy further supported the formation of supramolecular assemblies in solution. More specifically, systems containing **Ac-YF** and **Ac-CF** exhibited dense fibrillar assemblies, consistent with gelation and extended aggregation observed in TEM. In contrast, **Ac-Y** and **Ac-YD** displayed less organized structures. For **Ac-C** and **Ac-CD**, primarily spherical aggregates were observed

(Supplementary Fig. 30). These findings demonstrate that the the nucleophile and the overall peptide structure influence not only the sequence of covalent transformations but also the progression of supramolecular organization. Fluorescence and TEM analyses revealed that acylation typically drives early aggregation, while subsequent click reactions often induce disassembly. In cysteine-containing systems, this manifests as sol-gel-sol behavior and visible color changes, consistent with delayed CuAAC and persistent thioester intermediates. In contrast, tyrosine-based systems show minimal supramolecular reconfiguration due to the early and dominant onset of click chemistry. The assembly and reorganization of intermediates bearing distinct functional groups, such as esters, thioesters and triazole derivatives are thus closely linked to the temporal progression of covalent transformations.

**Sequence selection through nucleophile and azide competition**
Having explored the influence of cysteine and tyrosine residues individually on reaction sequence and self-assembly, we next examined a more complex system in which both nucleophiles are present within the same peptide substrate. The dipeptide **Ac-CY** contained both a thiol and a phenol offers simultaneous access to two acyl transfer pathways, in addition to the CuAAC reaction (Fig. 4a, Supplementary Figs. 31, 32). To ensure fair competition between the nucleophilic sites, the reaction was carried out using equimolar concentrations of **1-EP** (10 mM) and **Ac-CY** (10 mM), keeping the **azide** concentration identical to previous experiments (20 mM). This stoichiometry assisted to prevent the thiol from dominating the reaction, which can occur at higher peptide concentrations or under more basic conditions. Under these conditions, **Ac-CY-Thioester** (**7**) formation occurred first, reaching ~1.8 mM conversion within 2 h. In parallel, a smaller amount of **Ac-CY-Diester** (**8**) formed (~0.5 mM), while the CuAAC reaction remained suppressed during this period. We hypothesize that the free thiol of **Ac-CY** continues to sequester Cu(I), thereby limiting the

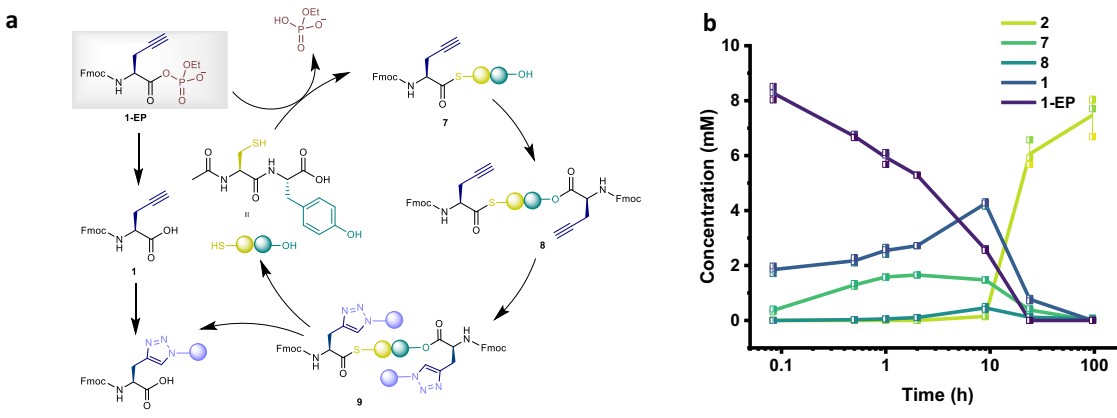

**Fig. 4 | Temporal programming of competing cascade reactions. a** Reaction network showing multiple competing pathways in the presence of **Ac-CY**, including sequential acyl transfer (thioester (**7**) and diester (**8**) formation) and CuAAC reactions (**2** and **9**). **b** Time-course concentration profile of the reaction mixture containing **1-EP** (10 mM), **azide** (20 mM), and **Ac-CY** (10 mM) in 0.2 M phosphate buffer at pH 8.0. Error bars represent the standard deviation from three independent experiments. A plot displaying the full product distribution including **2-EP** and **9** is provided in the Supplementary Information (Supplementary Fig. 32).

amount of active catalyst required to initiate CuAAC. Consequently, intermediate **8** forms prior to the onset of the click reaction. The onset of CuAAC was detected only after 9 h, as indicated by the formation of ~0.2 mM of click products (**2** and **9**). After 24 h, reaction product analysis indicated low concentrations of **Ac-CY-Diester-click** (**9**) (~0.4 mM), in stark contrast to the high concentration of the **Click** (**2**) (~6.6 mM), suggesting that the acylated intermediates were largely hydrolyzed before participating in CuAAC (Fig. 4c). These results reveal a distinct temporal separation between acyl transfer steps and the CuAAC reaction, even under equimolar conditions between nucleophile and activated monomer (Fig. 4b). Furthermore, this ordered progression demonstrates that two distinct acyl transfer events (**7**-thioester and **8**-diester formation) can be temporally resolved before the initiation of CuAAC.

Following the study of systems where two acyl transfers competed with a single azide, we next explored the reverse scenario: the competition between two azides for a single aminoacyl phosphate ester in addition to the acyl transfer reaction. Despite the widespread use of CuAAC, detailed studies of azide competition in an aqueous environment are relatively rare[77], due to the fast kinetics and typically high yield of the reaction. Here, we designed a system in which two azides, one aliphatic (6-azidohexanoic acid) and one aromatic (4-azidobenzoic acid) are present at equal concentration and compete for the same aminoacyl phosphate ester (Fig. 5a, Supplementary Fig. 33). In our first set of experiments, we used **1-EP** (10 mM), **azide** (10 mM), **aromatic azide** (10 mM), and **Ac-C** (40 mM) in 0.2 M phosphate buffer at pH 8.0 (Fig. 5). Under these conditions, acyl transfer proceeded efficiently, enabled thioester (**5a′**) formation of ~8.7 mM within 8 h. Given the rapid kinetics of CuAAC, minimal selectivity would be anticipated. However, the system displayed a clear preference: **6a′** reached about 4.8 mM, while **10a′** formed only around 1.7 mM (Fig. 5c). This suggests that structural differences between the azides influence their reactivity in the CuAAC step, with the aliphatic azide reacting more readily than the aromatic under these conditions, potentially reflecting electronic or solvation effects associated with the aromatic substituent. To determine whether this selectivity persists using a different nucleophile, we repeated the experiment using **Ac-Y** (40 mM) in place of **Ac-C**, with the same azide partners (Supplementary Figs. 34, 35). As in previous tyrosine-based systems, click reactions occurred before significant acyl transfer could take place. Within 1 h, the majority of the **1-EP** precursor had been consumed. Only trace amounts of hydrolysis product (**1**) or acylated intermediates were detected, indicating near-complete transformation via CuAAC. The system showed clear selectivity for the aliphatic azide across all

transformation pathways, with nearly double the conversion in clicked aminoacyl phosphate esters (~4.9 mM, **2-EP** vs. ~1.8 mM, **11-EP**) and ester-click (~1.2 mM, **4a** vs. ~0.5 mM, **12a**). After 72 h, the product distribution was dominated by **2** ( ~ 6.9 mM) and **11** ( ~ 3.2 mM) (Supplementary Fig. 35). Overall, these results show that azide structure significantly influences CuAAC selectivity, demonstrating how both peptide and azide features direct the order of multicomponent transformation pathways.

## Discussion

We report a one-pot chemical system in aqueous media, in which the sequence of covalent transformations is governed by the structure and reactivity of peptide-based nucleophiles reacting with aminoacyl phosphates bearing alkyne moieties. The phosphate groups act as solubility tags, enabling the dissolution of otherwise hydrophobic alkynes and peptide intermediates, thereby facilitating their participation in aqueous-phase transformations. Depending on the nucleophile, the system selectively prioritizes either acyl transfer or CuAAC, allowing reaction sequence to be directed through molecular design and self-assembly. In tyrosine-containing systems, the low nucleophilicity of phenolates causes CuAAC to outpace acyl transfer, preventing clear temporal separation and yielding mixed triazole-ester outcomes. Thiol-containing substrates, in contrast, rapidly generate thioesters and transiently sequester Cu(I), imposing an ordered sequence in which acyl transfer precedes CuAAC and self-assembly further reinforces this ordering. Incorporating both nucleophiles within a single peptide translates these principles into a three-step cascade, demonstrating that temporal programming can be embedded across multiple covalent transformations. Competition between aliphatic and aromatic azides reveals a consistent preference for aliphatic partners, indicating that azide structure also biases pathway selection. Macroscopic signatures such as gelation and color changes provide real-time readouts of these temporally regulated processes. These transformations proceed without enzymes or external triggers, relying instead on the intrinsic reactivity of nucleophiles, metal sequestration and phase behavior. The reconfiguration of soluble acyl phosphates[78] into esters or thioesters generates dynamic intermediates that modulate both the timing and outcome of subsequent transformations, including the click reaction. The ability to position multiple acyl transfer events upstream of irreversible steps might offer a modular approach to direct reaction cascades and sequences with intrinsic temporal order[79]. Looking ahead, we envision two complementary directions emerging from our study. First, the irreversible click step could, in principle, be

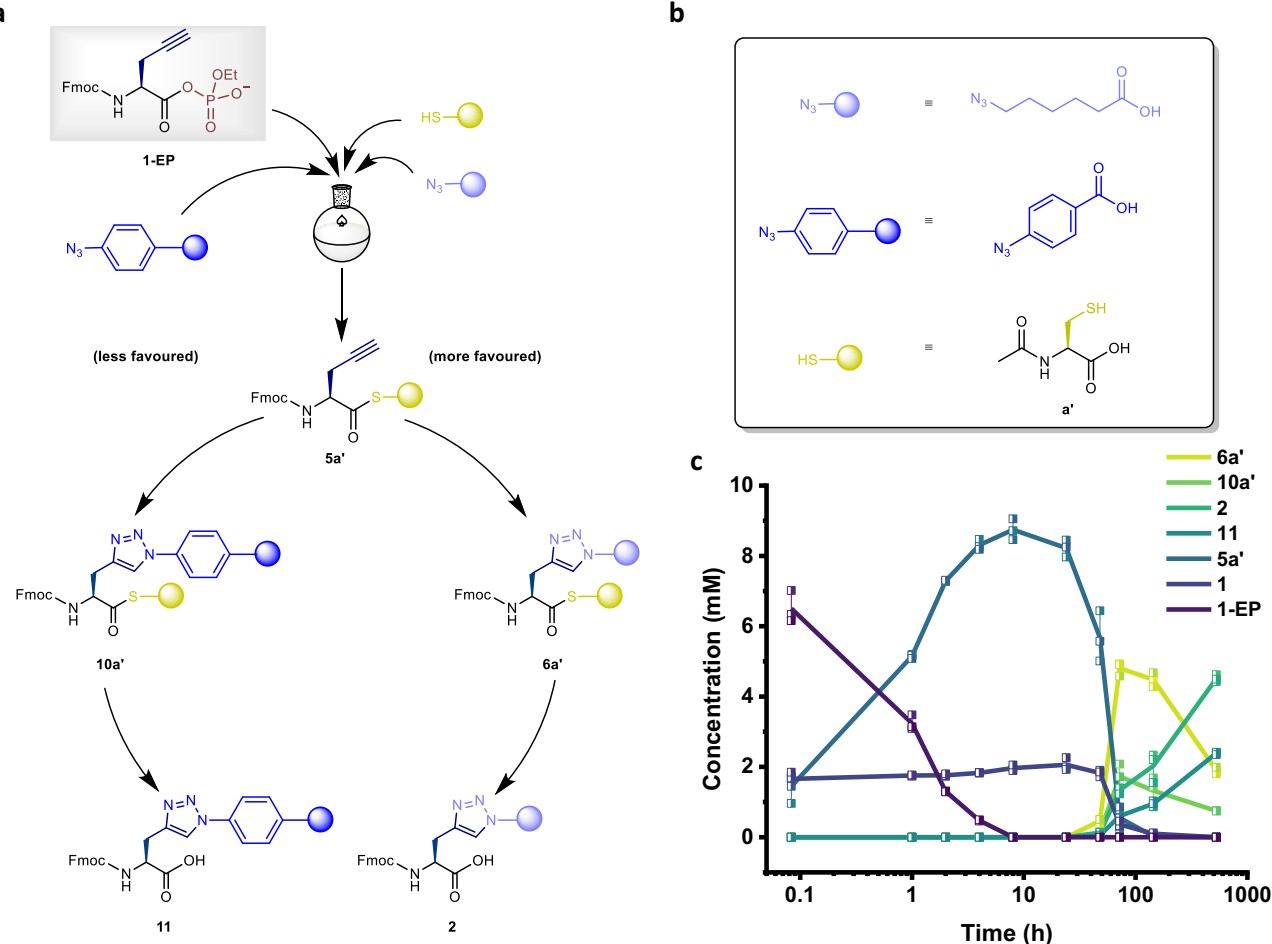

**Fig. 5 | Competitive CuAAC reactions between aliphatic and aromatic azides.** **a** Reaction network showing transformations in the presence of aliphatic and aromatic azides, starting with acyl transfer followed by competitive CuAAC processes (**6a', 10a', 2** and **11**). **b** Structures of the azides and peptides used. **c** Time-course concentration profile of the reaction mixture containing **1-EP** (10 mM), **azide** (10 mM), **aromatic azide** (10 mM) and **Ac-C** (40 mM) in 0.2 M phosphate buffer at pH 8.0. Error bars represent the standard deviation from three independent experiments.

replaced or complemented by dynamic reversible chemistries, such as imine formation or thiol-disulfide exchange, which in turn may introduce adaptability, error correction, and feedback into related reaction networks. Second, the exceptional robustness and orthogonality of CuAAC provide a reliable handle for generating diverse, functionalized aminoacyl phosphate intermediates, which may enable selective and potentially orthogonal acyl-transfer processes even within more complex chemical environments.

## Methods

### Materials

All reagents were purchased from Sigma-Aldrich and Carl Roth and were used without any further purification. Fmoc-amino acids were purchased from Iris Biotech, Carbolution and ABCR. The cysteine and tyrosine amino acid derivatives and peptides were purchased from Carbolution, Genscript and BLD pharm. The azides were purchased from ABCR and TCI. Preparative RP-MPLC was performed using an automated Interchim-puriFlash system.

### Sample preparation

All cysteine-containing peptide libraries were prepared in 0.2 M sodium phosphate-buffer at pH 8.0, while tyrosine-containing peptide libraries were prepared in 0.2 M bicarbonate buffer at pH 10.0. In all cases, Fmoc-aminoacyl phosphate ester was used at a final concentration of 10 mM in 0.5 mL of buffer. Peptides and azides were

added at varying concentrations (10–40 mM), depending on the specific library. For CuAAC initiation, 1–4 µL of 1 M sodium ascorbate solution and 5–20 µL of 100 mM $CuSO_4 \cdot 5H_2O$ were added, proportional to the amount of azide used (10–40 mM) i.e., for 20 mM azide, 2 µL of sodium ascorbate solution and 10 µL of $CuSO_4 \cdot 5H_2O$ solution were used. Peptides and azides were pre-weighed in vials and dissolved in buffer, followed by the sequential addition of sodium ascorbate and $CuSO_4 \cdot 5H_2O$. The resulting solution was then transferred quickly to a second vial containing the pre-weighed Fmoc-aminoacyl phosphate ester. The specific concentrations of components for each experiment are indicated in the corresponding figure legends.

### UPLC analysis

UPLC analyses were performed on a Waters Acquity UPLC H-Class Bio system, equipped with a photodiode array detector at a detection wavelength of 214 nm. Samples were injected on an Acquity UPLC CSH-C18 (150 × 2.1 mm) column, using UPLC-MS grade water eluent (A) and UPLC-MS grade acetonitrile eluent (B), which contained 0.1 % trifluoroacetic acid as the modifier. A flow rate of 0.3 ml min⁻¹ and a column temperature of 35 °C were applied. For all samples, vortex (15 s) and sonication (15 s) were performed prior to UPLC injection to ensure a homogeneous phase. Samples were prepared by taking 5 µl from the reaction vial and diluting (200 times) into $H_2O$ containing with 0.1% trifluoroacetic acid. Vortex and sonication (for 15 s) was furthermore applied after dilution, prior to UPLC injection.

## UPLC-MS analysis

Ultra-Performance Liquid Chromatography-Mass Spectrometry (UPLC-MS) experiments were performed on an Agilent 6546 LC/Q-TOF equipped with an infinity 1290 II in the LC section. We used the same UPLC column, as described in the UPLC analysis section above. The Q-TOF was equipped with a dual AJS ESI source. The experiments were conducted at a VCap voltage of 4000 V, a sheath gas temperature of 3000 C and a fragmentor voltage of 120 V. An internal reference was used.

## NMR

The $^1$H NMR spectra were recorded on a Bruker Advance Neo 500 MHz with broadband cryoprobe Prodigy. The $^{31}$P NMR spectra were recorded on Bruker 202 MHz spectrometers using $^1$H-broad band decoupling in the indicated deuterated solvent. Chemical shifts were reports as delta values from standard peaks.

## Transmission electron microscopy (TEM)

Imaging was performed at different time points of the reactions. Libraries were vortexed and sonicated prior to sample preparation for imaging. Small drops of the solution, the gel or the suspension were applied to a carbon-coated Cu grid for 30 s incubation. This was followed by two drops of water wash and one drop of 5 µl of 2% (w/v) uranyl acetate solution for 30 s was applied for staining. Excess solution was removed by blotting the grid with a piece of filter paper and left to air dry. Imaging was performed using a FEI Talos 120 C at 120 kV operating voltage. Images were taken using a CETA 16-megapixel camera.

## Confocal microscopy

Imaging was performed on a Zeiss LSM 880 confocal microscope system, equipped with a 63x oil immersion objective. Samples were mixed with 1 mM Nile red in DMSO to reach a final dye concentration of 1 µM and then transferred into ibidi µ-Slide 8-well Bioinert microplates. Samples were excited at 561 nm and emission signals were collected in the range of 575–630 nm.

## Fluorescence spectroscopy

Aminoacyl phosphate ester solutions (500 µL) were prepared in 0.2 M phosphate buffer (pH 8.0) for cysteine-containing peptides. The measurements for **5a´**, **5b´** and **5c´** were taken when **1-EP** was completely consumed. Following which **6a´**, **6b´** and **6c´** were measured when **5a´**, **5b´** and **5c´** were completely consumed by virtue of CuAAC. 50 µL from the solution were transferred into a 96 Well Microplate (black, 190 µm clear bottom, cycloolefin, cell culture treated, sterile with lid). Fluorescence emission spectra were recorded using an Agilent BioTek Synergy H1 spectrometer. Excitation was set at 280 nm, and emission was recorded from 300 to 700 nm at 25 °C.

## Data availability

All data generated or analyzed during this study are included in this article and its supplementary information files, and are available from the corresponding author. Source data has been provided with this paper. Source data are provided with this paper.

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

## Acknowledgements

The work was supported by the European Union (ERC-2023-StG grant, PhosphotoSupraChem, 101117240). Arti Sharma acknowledges support from the Deutsche Forschungsgemeinschaft (DFG, German Research Foundation) under project number 495280186. The authors thank Christoph Warth and Dr. Stefan Braukmüller for analytical support. Confocal imaging was performed at the Lighthouse Core Facility, supported in part by the Medical Faculty, University of Freiburg (Project Numbers 2023/A2-Fol; 2021/B3-Fol) and the DFG (Project Number 450392965).

## Author contributions

C.G.P. and D.B. conceived the research. C.G.P. supervised the overall project. D.B. synthesized the aminoacyl phosphate ester, performed the experiments and analyzed the data via UPLC-MS. A.S. performed the fluorescence experiments. K.D. and T.P. performed UPLC-MS measurements. L.S. performed the confocal experiments. R.T. performed the TEM experiments. C.G.P. and D.B. co-wrote the manuscript with the support of all the authors.

## Funding

## Competing interests

The authors declare no competing interests.
