## [Transparent Peer Review file · Nature Communications]

Pathway Selection between Click and Acyl Transfer Reactions Driven by Aminoacyl Phosphates

Corresponding Author: Dr Charalampos Pappas

Version 0:

Reviewer comments:

Reviewer #1

(Remarks to the Author)

Pappas and co-workers have shown how molecular structure, intrinsic reactivity and self-assembly can together direct the order, timing and rate of reactions in peptide based systems mimicking biological systems with self-regulated reaction sequences. They have studied chemical systems consisting alkyne amino acyl phosphate esters, azides and peptides with cysteine/tyrosine or both to make thioester and Copper catalyzed click reaction. I find the work very interesting and timely, clearly signaling how reaction and self-assembly can modulate each other to give complex outcomes via unexpected pathways. However, I have few minor comments which will need attention from the authors.

- 1) As the authors mention, "Biological functions are encoded in temporally ordered reaction sequences". Most of the times, these are dynamic and reversible, as well. Now, click reaction are robust and versatile but lack this dynamicity. Can the authors comment on why this chemistry was coupled with their amino acyl phosphate chemistry, and which could be other viable options.
- 2) The aromatic azide is mentioned early in the manuscript but the experiments involving it are not discussed until the very end. I would suggest to not mention it until later to avoid confusion to the readers.
- 3) The labels Pathway I and Pathway II in the figure 1, could be shifted closer to the respective reaction scheme to clearly show which is which. Currently they lie confusingly close to the central reaction pathway.
- 4) The authors mention that the -ve Aspartic acid (-COO-) might influence the reactivity by repulsion with the phosphate groups. Can they elaborate on this..is it due to the change in the inherent molecular reactivity, due to the effect on self-assembly propensity, or something else. (line 120, page 3).
- 5) The authors mention that CuAAC is kinetically favoured (line 124, page 4), can they elaborate or discuss the reason for this.
- 6) In figure 2, and throughout the MS, it would be great to have same symbols and colors in different graphs for the same chemical species. This would greatly help in quickly following them through different panels.
- 7) I suggest, it would be best to keep the same style of naming chemical species throughout instead of using different terms interchangeably. For instance, using nucleophile in place of amino acid/names in caption for figure 2, click product instead of 2 (line 286) etc.
- 8) The authors mention the role of thiols on chelating the Cu(I). Can they comment on the likelihood of -COO- doing the same?
- 9) The authors mention that the trapping of reactive species inside the assemblies can cause kinetic mismatch (line 147, page 5). Do they observe the hydrogel changing over time. As even though CuAAC is predominant in the solution, but over time it could cause leeching from the fibers to replace the consumed species from the solutions.
- 10) When the authors mention weak gelation (lines 168-169, page 5), can they suggest which species gels, mention what is the concentration range at which gelation is observed and around what time since the start of the reaction. Similarly, for the other instances where gelation is mentioned in the manuscript.
- 11) In the figure 3 and similar microscopy images, it would be informative to have vial images and mentions of gels/not gel/precipitate like descriptions allowing to connect microscopic to macroscopic appearance.
- 12) In the discussions related to the Figure 4. Can the authors comment/rationalize the absence of click product from 7 before 8 forms, which could also create a direct bypass to 2 after hydrolysis.
- 13) Can the authors comment on the similarity of the ratios of aliphatic to aromatic click products obtained for cysteine and tyrosine based systems (4.9:1.8) Specifically, the absence of any effect from the self-assembly of Ac-C based thioester [assuming it gels (not mentioned)].
- 14) Please check the typo in line 307- 10a or 10a'. Similarly, other typos for example line 297, page 8.

Reviewer #2

(Remarks to the Author)

The manuscript by Bhattacharjee et al. explores an abiotic aqueous system where aminoacyl phosphate esters undergo temporally controlled covalent transformations guided by peptide nucleophiles. Cysteine residues enable selective thioester formation that delays CuAAC, achieving sequential reactions through thiol-copper coordination and self-assembly mediated regulation. Embedding both cysteine and tyrosine in one peptide yields a defined three-step sequence, illustrating how reactivity and supramolecular context can impose temporal order in synthetic chemical reaction networks.

Overall, the work is supported by extensive characterizations and appropriate conclusions. I recommend the article for publication after addressing the following minor points mentioned below.

Points:

1. In the case of Ac-YD, due to the negatively charged side chain of aspartic acid, electrostatic repulsion with the phosphate moiety is expected to result in lower formation of 3b compared to 3a (in case of Ac-Y). However, the results show that 3a is not detectable at all, while some amount of 3b is formed. It would be good to add some discussion about this point.
2. Why was pH 8.0 used for the cysteine-containing peptides? Apart from reducing the hydrolysis of 1-EP, does the buffer have any effect on the azide reactivity, since compound 1 remained freely available during early time points without undergoing the click reaction?
3. ".....the onset of CuAAC was delayed relative to both Ac-C and Ac-CD." Since the X-axis is plotted on a log scale, the time progression is less intuitive to interpret.
4. "The persistent variation in timing among Ac-C, Ac-CD and Ac-CF instead suggests that self-assembly further modulates the temporal profile....." When comparing 40 mM Ac-CF with 20 mM Ac-CF, the delayed click reaction (formation of 6c') could be due to the lower generation of 5c', rather than solely the effect of self-assembly. Please discuss this point.
5. In my opinion, the narration would be much easier to understand if the microscopic investigation were presented in parallel with the product distribution of the different systems. The authors may consider revising it accordingly although this is just a suggestion.
6. SI Fig. 16: The different fluorescence spectra are shown to correspond different species (such as 5a', 6a') but how this was done is not clear from the method section.
7. SI Fig. 22 & 23: The 72 h TEM images appear to be the same for both cases, although the experiments seem to be different. Please correct this.
8. Figure 4: Are there any traces of mono-click product formation?
9. The conclusion should be emphasised further to better convey the importance of the study.

Version 1:

Reviewer comments:

Reviewer #1

(Remarks to the Author)

I am now satisfied with the changes made by the authors. I thank and congratulate the authors on their extensive effort in revising this MS.

Reviewer #2

(Remarks to the Author)

The manuscript has been aptly revised and all the raised points have been addressed. I recommend acceptance in its present form.

Dibyendu Das

Reviewer #1 (Remarks to the Author):

Pappas and co-workers have shown how molecular structure, intrinsic reactivity and self-assembly can together direct the order, timing and rate of reactions in peptide-based systems mimicking biological systems with self-regulated reaction sequences. They have studied chemical systems consisting alkyne amino acyl phosphate esters, azides and peptides with cysteine/tyrosine or both to make thioester and Copper catalyzed click reaction. I find the work very interesting and timely, clearly signaling how reaction and self-assembly can modulate each other to give complex outcomes via unexpected pathways. However, I have few minor comments which will need attention from the authors. 1) As the authors mention, “Biological functions are encoded in temporally ordered reaction sequences”. Most of the times, these are dynamic and reversible, as well. Now, click reaction are robust and versatile but lack this dynamicity. Can the authors comment on why this chemistry was coupled with their amino acyl phosphate chemistry, and which could be other viable options.

Response: We thank the reviewer for the comment. We fully agree that biological reaction networks often rely on dynamic and reversible transformations, whereas the CuAAC click reaction is essentially irreversible and therefore lacks this inherent dynamicity. In the present work, however, our primary objective was to identify two mutually orthogonal reaction pathways that could operate in parallel within a single mixture without cross-reactivity. This design enabled us to test whether supramolecular assembly can impose temporal ordering and rate modulation on reactions that are otherwise independent. For this reason, we paired the acyl-transfer chemistry of aminoacyl phosphate esters with the CuAAC click reaction, taking advantage of their high compatibility and orthogonality under aqueous conditions. We also appreciate the reviewer’s comment on other dynamic chemistries. Indeed, this is an exciting direction for us, and we are now actively exploring the integration of acyl phosphate chemistry with reversible reactions, including imine formation and thiol–disulfide exchange. Such systems hold promise for introducing error-correction mechanisms, adaptive behavior, and more biologically inspired regulation. We have updated the manuscript to clarify our rationale for the reaction choice in this study and to highlight how reversible chemistries could be incorporated in future work in the conclusion section.

We have introduced the following text in the manuscript: “Despite its lack of inherent dynamicity relative to biological reaction networks, CuAAC offers a uniquely robust and orthogonal transformation that enables us to isolate the influence of supramolecular organization on chemical reactivity. Variations in local microenvironment can regulate site accessibility and thereby govern the timing and ordering of covalent processes. Accordingly, we aim to establish systems in which reaction sequences are dictated by intrinsic molecular architecture and environmental cues rather than by external control.”

We have also modified the conclusion section as follows: “We report a one-pot chemical system in aqueous media, in which the sequence of covalent transformations is governed by the structure and reactivity of peptide-based nucleophiles reacting with aminoacyl phosphates bearing alkyne moieties. The phosphate groups act as solubility tags, enabling the dissolution of otherwise hydrophobic alkynes and peptide intermediates, thereby facilitating their participation in aqueous-phase transformations. Depending on the nucleophile, the system selectively prioritizes either acyl transfer or CuAAC, allowing reaction sequence to be directed through molecular design and self-assembly. In tyrosine-containing systems, the low nucleophilicity of phenolates causes CuAAC to outpace acyl transfer, preventing clear temporal separation and yielding mixed triazole-ester outcomes. Thiol-containing substrates in contrast, rapidly generate thioesters and transiently sequester Cu(I), imposing an ordered sequence in which acyl transfer precedes CuAAC and self-assembly further reinforces this ordering. Incorporating both nucleophiles within a single peptide translates these principles into a three-step cascade, demonstrating that temporal programming can be embedded across multiple covalent transformations. Competition

between aliphatic and aromatic azides reveals a consistent preference for aliphatic partners, indicating that azide structure also biases pathway selection. Macroscopic signatures such as gelation and color changes provide real-time readouts of these temporally regulated processes. These transformations proceed without enzymes or external triggers, relying instead on the intrinsic reactivity of nucleophiles, metal sequestration and phase behavior. The reconfiguration of soluble acyl phosphates into esters or thioesters generates dynamic intermediates that modulate both the timing and outcome of subsequent transformations, including the click reaction. The ability to position multiple acyl transfer events upstream of irreversible steps might offer a modular approach to direct reaction cascades and sequences with intrinsic temporal order. Looking ahead, we envision two complementary directions emerging from our study. First, the irreversible click step could, in principle, be replaced or complemented by dynamic reversible chemistries, such as imine formation or thiol-disulfide exchange, which in turn may introduce error correction and feedback into reaction networks. Second, the exceptional robustness and orthogonality of CuAAC provide a reliable handle for generating diverse, functionalized aminoacyl phosphate intermediates, which may enable selective and potentially orthogonal acyl-transfer processes even within more complex chemical environments.”

2) The aromatic azide is mentioned early in the manuscript but the experiments involving it are not discussed until the very end. I would suggest to not mention it until later to avoid confusion to the readers.

Response: We have removed the early mention of the aromatic azide from the Results section. The term is now introduced later in the manuscript, alongside the experiments involving multiple azides to drive pathway selection.

3) The labels Pathway I and Pathway II in the figure 1, could be shifted closer to the respective reaction scheme to clearly show which is which. Currently they lie confusingly close to the central reaction pathway.

Response: We agree with the reviewer that the position of pathway I and II may be confusing in Fig.1 and have been shifted closer to their respective reaction scheme in the updated manuscript.

4) The authors mention that the –ve Aspartic acid (–COO–) might influence the reactivity by repulsion with the phosphate groups. Can they elaborate on this..is it due to the change in the inherent molecular reactivity, due to the effect on self-assembly propensity, or something else. (line 120, page 3).

Response: We thank the reviewer for raising this point. The negatively charged aspartate side chain can influence both the intrinsic molecular reactivity of the aminoacyl phosphate ester and the self-assembly propensity of the peptide. As previously reported (Nat. Commun. 16, 1306, doi:10.1038/s41467-025-56432-6 (2025), electrostatic repulsion between the aspartate carboxylate and the phosphate group can reduce the efficiency of acyl transfer, and this decreased reactivity often correlates with diminished assembly. These repulsive effects can be further amplified when the reactive species participate in supramolecular assemblies, whether in the form of the acyl phosphate, an ester, or a thioester. In such environments, higher local charge density affects both reaction and assembly pathways. In the present study, we examined Ac-YD and Ac-CD to assess how introducing a negatively charged residue modifies pathway selection. For Ac-YD, the acyl transfer was not competitive with CuAAC, and the solution remained clear without gelation, in contrast to hydrophobic systems such as Ac-YF. In the case of Ac-CD, the acyl-transfer yields were comparable to other cysteine-containing peptides; however, the absence of gelation reduced kinetic constraints and allowed CuAAC to initiate earlier. Our purpose in including aspartic acid in the nucleophiles was to probe how changes in polarity

could modulate the balance between reactivity and assembly relative to the unmodified systems (Ac-Y and Ac-C).

We have updated the following text: “Taking into account the negatively charged side chain of aspartic acid, we hypothesized that electrostatic repulsion with the phosphate moiety could influence both intrinsic reactivity and supramolecular assembly. □ Supporting this hypothesis, only a small amount of Ac-YD-ester (3b) (~0.1 mM) accumulated within the first 30 minutes before being rapidly converted into the Ac-YD-ester–click product (4b). This behaviour indicates that, in this system, acyl transfer proceeds more slowly than CuAAC. It further suggests that in the corresponding tyrosine system (Ac-Y), even if the ester intermediate 3a forms transiently, it is likely consumed immediately to generate the click product 4a, preventing its accumulation. Under these conditions, 1-EP is first converted via CuAAC to ~5 mM of 2-EP, followed by esterification to give 4b, which ultimately undergoes hydrolysis to yield 2. Together, these observations show that although acyl transfer and CuAAC can occur concurrently in one pot, CuAAC proceeds more rapidly when tyrosine is the nucleophile, owing to the relatively weak nucleophilicity of the phenolic side chain. Consequently, the faster kinetics of CuAAC limit the buildup of early acyl-transfer intermediates, which may also escape detection due to rapid conversion or hydrolysis on experimentally accessible timescales.”

5) The authors mention that CuAAC is kinetically favoured (line 124, page 4), can they elaborate or discuss the reason for this.

Response: In the manuscript, “kinetically favoured” refers to the observation that CuAAC proceeds significantly faster than the acyl-transfer reaction under the conditions tested, particularly for tyrosine-based nucleophiles (Ac-Y and Ac-YD). As shown in Fig. 2a and Fig. 2b, 1-EP is fully converted to the triazole product 2-EP before acyl transfer is complete. Although the reactions were performed in bicarbonate buffer at pH 10, tyrosine possesses a relatively weak nucleophile (phenolate) compared to cysteine (thiolate), leading to inherently slower acyl-transfer kinetics. This behavior allows the CuAAC reaction to dominate the early stages of the reaction sequence. In contrast, cysteine-containing systems exhibit much faster acyl transfer, which can transiently outcompete CuAAC.

We have updated the following text in the manuscript: “This kinetic mismatch allows acyl transfer to proceed first, even though CuAAC remains the faster reaction in solution, effectively reversing the reaction sequence observed in unassembled systems. Nevertheless, multiple species, including 3c, 2 and Ac-YF-Ester-click (4c) coexist over an extended period of 7 days, indicating that self-assembly enables partial but not complete temporal separation between acylation and click pathways.”

6) In figure 2, and throughout the MS, it would be great to have same symbols and colors in different graphs for the same chemical species. This would greatly help in quickly following them through different panels.

Response: We have maintained consistent symbols and colors for the same species within Figure 2. However, due to variations in the number of species displayed across different figures, it is not always feasible to maintain identical symbols and colors throughout the manuscript.

7) I suggest, it would be best to keep the same style of naming chemical species throughout instead of using different terms interchangeably. For instance, using nucleophile in place of amino acid/names in caption for figure 2, click product instead of 2 (line 286) etc.

Response: We thank the reviewer for pointing out this. It has been updated in the revised manuscript.

8) The authors mention the role of thiols on chelating the Cu(I). Can they comment on the likelihood of $-\text{COO}^-$ doing the same?

Response: We thank the reviewer for this question. Thiols readily coordinate Cu(I) owing to their high nucleophilicity and strong affinity for metals. In contrast, phenolic oxygen atoms are significantly less nucleophilic and do not effectively chelate Cu(I), as also reflected in our tyrosine-based systems. For carboxylate groups ($-\text{COO}^-$), the negative charge is delocalized over two oxygen atoms, reducing their ability to act as strong, directional ligands for Cu(I). If carboxylate–Cu(I) coordination were significant under our conditions, we would expect to see suppression or delay of CuAAC in the tyrosine derivatives (C-terminus carboxylates) or in experiments involving an inert thiol. However, no such effects were observed. This suggests that $-\text{COO}^-$ groups do not sequester Cu(I) in our system and therefore do not influence CuAAC kinetics.

9) The authors mention that the trapping of reactive species inside the assemblies can cause kinetic mismatch (line 147, page 5). Do they observe the hydrogel changing over time. As even though CuAAC is predominant in the solution, but over time it could cause leeching from the fibers to replace the consumed species from the solutions.

Response: We thank the reviewer for the question. In our study, the temporal behaviour is governed by a combination of thiol–Cu(I) sequestration and self-assembly. For tyrosine derivatives, CuAAC dominates due to the poor nucleophilicity of tyrosine, and the resulting triazole formation produces more polar species that do not support gelation (with the exception of Ac-YF). In these systems, both reactions proceed in solution and no time-dependent changes in morphology were observed. In contrast, cysteine-containing peptides (Ac-C and Ac-CF) form hydrogels, where thioester-containing intermediates are initially dominant. We observed that these gels weaken progressively as the click reaction initiates and proceeds, ultimately transitioning from a gel to a solution. We hypothesize that triazole formation increases the polarity of the acyl-transfer products, thereby reducing their propensity to remain within the hydrophobic fibrillar network. This behaviour supports the idea that supramolecular assemblies can transiently trap reactive species, but that consumption of these species through CuAAC ultimately drives their release and leads to dissolution of the gel.

Furthermore, we have provided time-dependent digital photographs for Ac-C illustrating its transformation from sol to gel and back to sol (Supplementary Fig. 19).

10) When the authors mention weak gelation (lines 168-169, page 5), can they suggest which species gels, mention what is the concentration range at which gelation is observed and around what time since the start of the reaction. Similarly, for the other instances where gelation is mentioned in the manuscript.

Response: In our experiments, we monitored the assembly behavior of the aminoacyl phosphate ester prior to acyl transfer and found that although it forms microscopic aggregates (Supplementary Fig. 21), it always remains a clear solution under the conditions tested. Gelation only appears when the aminoacyl phosphate ester is combined with nucleophilic amino acids or peptides and an azide. When Ac-C was used, we observed weak gelation, which we attribute to the formation of thioester intermediate, which was the dominant product. Gelation was also observed in systems containing Ac-CF and Ac-YF, consistent with their greater hydrophobicity and assembly propensity. We now provide the concentration and time that requires for gelation in the revised manuscript and in the supporting information.

For 1-EP (10 mM) in 0.2 M phosphate buffer at pH 8.0:

For 1-EP, azide and Ac-C in 0.2 M phosphate buffer at pH 8.0. They have been used in 1:2:4 ratio respectively:

1-EP Concentration	Gelation time	Thioester (5a') concentration during gelation
2 mM	-	-
5 mM	-	-
10 mM	10 h	8.22 mM
20 mM	4 h	8.17 mM

For **1-EP**, **azide** and **Ac-CF** in 0.2 M phosphate buffer at pH 8.0. They have been used in 1:2:4 ratio respectively:

1-EP Concentration	Gelation time	Thioester (5c') concentration during gelation
2 mM	-	-
5 mM	-	-
10 mM	30 min	7.08 mM
20 mM	10 min	11.52 mM

For **1-EP**, **azide** and **Ac-YF** in 0.2 M bicarbonate buffer at pH 10.0. They have been used in 1:2:4 ratio respectively:

1-EP Concentration	Gelation time	Ester (3c) concentration during gelation
2 mM	-	-
5 mM	-	-
10 mM	10 min	3.11 mM
20 mM	<1 min	3.96 mM

Gelation was observed only for 10 mM and 20 mM concentration ratio. No gelation was observed for 2 mM and 5 mM. The respective time for gelation and the thioester concentration at the time of gelation are provided above for **Ac-C**, **Ac-CF** and **Ac-YF**.

The time-dependent vial images and the gelation concentration has also been provided in the supporting information (Supplementary Fig. 14-17)

11) In the figure 3 and similar microscopy images, it would be informative to have vial images and mentions of gels/not gel/precipitate like descriptions allowing to connect microscopic to macroscopic appearance.

Response: We agree that correlating morphology with macroscopic appearance enhances interpretability, and we have now incorporated vial photographs alongside the corresponding microscopy images in Figure 3 which is given below.

For Ac-C, the sample initially appeared as a clear solution, although microscopic spherical aggregates were present and coincided with thioester formation. After 24 h, the system developed into a weak gel, reflected by denser aggregates microscopically. By 72 h, when click products predominated, the material reverted to a solution.

For Ac-CD, the sample remained macroscopically clear throughout, despite persistent microscopic aggregates at all measured time points.

For Ac-CF, gelation occurred within 30 min. After seven days, as click products accumulated, the gel weakened and transitioned to a solution.

For tyrosine derivatives (Ac-Y and Ac-YD), the samples were always translucent solutions macroscopically, even though TEM revealed aggregate formation.

We have added the relevant vial photographs for Figure 3 in the revised manuscript. While we considered including vial images for all additional TEM data in the Supplementary Information, we think this would lead to redundancy because these experiments closely parallel those in Figure 3. Nonetheless, vial photographs for Ac-YF have been added to the SI, as in this case, they offer additional insights.

12) In the discussions related to the Figure 4. Can the authors comment/rationalize the absence of click product from 7 before 8 forms, which could also create a direct bypass to 2 after hydrolysis.

Response: We thank the reviewer for the comment. In this system, the temporal control is governed primarily by thiol–Cu(I) sequestration and secondarily by supramolecular assembly. When Ac-CY is used as the nucleophile, the thiol of the cysteine residue reacts first to generate intermediate 7. The second acyl-transfer event at the phenolic site then occurs before CuAAC is initiated. We attribute this ordering to the continued ability of the free thiol in Ac-CY to coordinate Cu(I), thereby temporarily reducing the pool of catalytically active copper available to trigger the click reaction. As a result, intermediate 8 forms prior to any detectable CuAAC, preventing the early appearance of click-derived products from 7 and eliminating a bypass pathway to 2 via hydrolysis.

We have added the following sentence in the revised manuscript: “We hypothesize that the free thiol of Ac-CY continues to sequester Cu(I), thereby limiting the amount of active catalyst required to initiate CuAAC. Consequently, intermediate 8 forms prior to the onset of the click reaction.”

13) Can the authors comment on the similarity of the ratios of aliphatic to aromatic click products obtained for cysteine and tyrosine-based systems (4.9:1.8) Specifically, the absence of any effect from the self-assembly of Ac-C based thioester [assuming it gels (not mentioned)].

Response: We thank the reviewer for highlighting this point. In the Ac-C system, the reported ratio reflects the relative formation of the two click products 6a' and 10a' (4.8:1.7), both arising from the same thioester intermediate (5a'). In this case, only weak gelation is observed, and as the CuAAC reaction progresses, the system transitions back to a solution phase. The transient and limited nature of the assembly likely minimizes any strong bias on the relative azide incorporation. For the Ac-Y system, 1-EP is reconfigured into 2-EP and 11-EP, yielding a ratio of 4.9:1.8. As correctly noted by the reviewer, the similarity lies in comparing the ratios of (6a':10a') for Ac-C and (2-EP:11-EP) for Ac-Y. Since the same pair of azides was used in both cases, this similarity suggests an inherent preference between the aliphatic and aromatic azides that dominates over assembly effects in these systems. However, for a complete comparison of aliphatic versus aromatic click reactivity in the Ac-Y system, the contributions from the ester-click products (4a and 12a) should also be considered. Including these species adjusts the overall ratio for Ac-Y to 6.1:2.4, further underscoring that azide structure plays a primary role in governing product distribution.

14) Please check the typo in line 307- 10a or 10a'. Similarly, other typos for example line 297, page 8.

Response: We thank the reviewer for pointing out the typos. It has been corrected in the revised manuscript.

Reviewer #2 (Remarks to the Author):

The manuscript by Bhattacharjee et al. explores an abiotic aqueous system where aminoacyl phosphate esters undergo temporally controlled covalent transformations guided by peptide nucleophiles. Cysteine residues enable selective thioester formation that delays CuAAC, achieving sequential reactions through thiol-copper coordination and self-assembly mediated regulation. Embedding both cysteine and tyrosine in one peptide yields a defined three-step sequence, illustrating how reactivity and supramolecular context can impose temporal order in synthetic chemical reaction networks. Overall, the work is supported by extensive characterizations and appropriate conclusions. I recommend the article for publication after addressing the following minor points mentioned below.

Response: We thank the reviewer for the positive assessment of our work and for recognizing the significance of temporally controlled covalent transformations in abiotic systems. We are grateful for the positive recommendation and have carefully addressed all comments point-by-point below.

Points:

1. In the case of Ac-YD, due to the negatively charged side chain of aspartic acid, electrostatic repulsion with the phosphate moiety is expected to result in lower formation of 3b compared to 3a (in case of Ac-Y). However, the results show that 3a is not detectable at all, while some amount of 3b is formed. It would be good to add some discussion about this point.

Response: We thank the reviewer for raising this point. Our intention was not to imply that 3a is not formed in the Ac-Y system, but rather that it does not accumulate to a detectable level under our experimental conditions. In tyrosine-containing systems, acyl transfer and CuAAC proceed in parallel; however, because CuAAC is significantly faster, any 3a formed is rapidly converted into the ester-click product 4a and therefore cannot be readily detected. By the time acyl transfer would otherwise be complete, the aminoacyl phosphate (1-EP) has already been largely transformed into the click product 2-EP. We also note that we cannot exclude the possibility that 3a forms transiently and undergoes rapid conversion or hydrolysis on timescales that are not resolvable with the chromatographic techniques employed. The detection of a small but measurable amount of 3b (~0.1 mM) in the Ac-YD system supports this interpretation, as the slower kinetics in this case allow partial accumulation of the ester intermediate. This behaviour is further corroborated by experiments with Ac-YF. We have clarified these points in the revised manuscript.

Also, we have updated the following text in the manuscript: "Taking into account the negatively charged side chain of aspartic acid, we hypothesized that electrostatic repulsion with the phosphate moiety could influence both intrinsic reactivity and supramolecular assembly. Supporting this hypothesis, only a small amount of Ac-YD-ester (3b) (~0.1 mM) accumulated within the first 30 minutes before being rapidly converted into the Ac-YD-ester-click product (4b). This behaviour indicates that, in this system, acyl transfer proceeds more slowly than CuAAC. It further suggests that in the corresponding tyrosine system (Ac-Y), even if the ester intermediate 3a forms transiently, it is likely consumed immediately to generate the click product 4a, preventing its accumulation. Under these conditions, 1-EP is first converted via CuAAC to ~5 mM of 2-EP, followed by esterification to give 4b, which ultimately undergoes hydrolysis to yield 2. Together, these observations show that although acyl transfer and CuAAC can occur concurrently in one pot, CuAAC proceeds more rapidly when tyrosine is the nucleophile, owing to the relatively weak nucleophilicity of the phenolic side chain. Consequently, the faster kinetics of CuAAC limit the buildup of early acyl-transfer intermediates, which may also escape detection due to rapid conversion or hydrolysis on experimentally accessible timescales."

2. Why was pH 8.0 used for the cysteine-containing peptides? Apart from reducing the hydrolysis of 1-EP, does the buffer have any effect on the azide reactivity, since compound 1 remained freely available during early time points without undergoing the click reaction?

Response: We thank the reviewer for the question. pH 8.0 was selected for the cysteine-containing peptides to balance two competing considerations: minimizing background hydrolysis of the aminoacyl phosphate ester while maintaining sufficient nucleophilic activity of the thiol. Because we did not use an external ligand to stabilize Cu(I) in the CuAAC reaction, the buffer conditions also needed to avoid coordinating with copper or promoting its precipitation. Higher pH values increase the likelihood of forming $\text{Cu}(\text{OH})_2$, whereas lower pH would reduce thiolate availability. Notably, pH 7–9 is commonly employed for CuAAC in aqueous systems (Bioconjugate Chem. 29, 686–701, doi:10.1021/acs.bioconjchem.7b00633 (2018)). Although compound 1 remained freely available at early time points, the Cu(I) required to initiate the click reaction was transiently sequestered by the cysteine thiol, which delays the formation of catalytically active copper species. As a result, CuAAC does not proceed immediately despite the presence of unreacted 1.

We have updated the manuscript as follows: “The buffer conditions were selected to minimize hydrolysis of the phosphate ester while preserving the high nucleophilicity of thiols. Moreover, previous studies have reported that CuAAC proceeds efficiently in aqueous media at pH 7-9.⁷⁶ Accordingly, pH 8.0 provides an optimal balance between limiting ester hydrolysis, maintaining thiol nucleophilicity, and enabling efficient CuAAC. However, for the tyrosine derivatives 0.2 M bicarbonate buffer was used keeping in mind the nucleophilicity of phenolic derivatives.”

3. “.....the onset of CuAAC was delayed relative to both Ac-C and Ac-CD.” Since the X-axis is plotted on a log scale, the time progression is less intuitive to interpret.

Response: We appreciate the reviewer's point that a logarithmic time axis can make the progression of events less intuitive. Because the different systems exhibit reaction events on markedly different timescales, we chose a log scale to allow direct visual comparison across all systems within a single plot. We have now added explicit time ranges for the key event transitions in the main text of the revised manuscript, allowing readers to interpret the kinetic progression more readily while retaining the comparative value of the log-scale plot.

4. “The persistent variation in timing among Ac-C, Ac-CD and Ac-CF instead suggests that self-assembly further modulates the temporal profile.....” When comparing 40 mM Ac-CF with 20 mM Ac-CF, the delayed click reaction (formation of 6c') could be due to the lower generation of 5c', rather than solely the effect of self-assembly. Please discuss this point.

Response: We thank the reviewer for this observation. The delayed onset of CuAAC is governed primarily by thiol–Cu(I) sequestration, with self-assembly contributing as a secondary factor. When comparing 40 mM and 20 mM Ac-CF, we agree that the lower level of thioester intermediate 5c' at 20 mM would reduce the extent of assembly, which could in turn influence the temporal profile. However, the change in concentration also directly affects the amount of free thiol available to sequester Cu(I). At 20 mM, the reduced thiol concentration diminishes the extent and duration of Cu(I) sequestration, enabling earlier release of catalytically active copper and thus an earlier initiation of the CuAAC reaction. Therefore, both factors, the reduced formation of 5c' and the lower thiol-mediated metal sequestration contribute to the differences observed between the two concentrations.

This aspect has been highlighted in the manuscript through the following text: “To evaluate the impact of concentration, we repeated the experiments using a lower peptide (20 mM) and a higher azide concentration (40 mM) (Supplementary Fig. 8, 10). Under these conditions, both the Ac-CD and Ac-CF systems showed an earlier onset of CuAAC following acyl transfer, similar to Ac-C (Fig. 2e). However, the temporal profiles of the systems remained distinct. If copper sequestration were the sole factor governing temporal control, one would expect identical delays across all cysteine-containing substrates. The persistent variation in timing among Ac-C, Ac-CD and Ac-CF instead suggests that self-assembly further modulates the temporal profile, amplifying the separation between thioester formation and CuAAC, by influencing the accessibility and diffusion of reactants. Additional support for this conclusion came from co-solvent experiments. We hypothesized that by diminishing hydrophobic interactions in the presence of organic solvent, the delay between acyl transfer and CuAAC could be affected. When reactions were conducted in a 1:1 mixture of DMSO and phosphate buffer (10 mM 1-EP, 20 mM azide, 40 mM Ac-C / Ac-CF; Supplementary Fig. 11,12), the delay between thioester formation and click reaction was significantly reduced. To further probe the role of aggregation, we repeated the experiments at lower reactant concentrations (5 mM 1-EP, 10 mM azide, 20 mM Ac-C; Supplementary Fig. 13), under which no gelation was observed. Interestingly, the same reaction sequence was maintained, with acyl transfer still preceding CuAAC. More broadly, across different systems, concentration-dependent experiments and macroscopic observations indicate that gelation emerges only above a threshold phosphate ester concentration and coincides with the accumulation of acyl-transfer intermediates (Supplementary Fig. 14-17), whereas lower concentrations favored homogeneous solutions. The hypothesis that thiol-mediated metal sequestration delays the onset of CuAAC, was further confirmed from experiments with excess copper sulfate (Supplementary Fig. 18). Under these conditions, CuAAC proceeded exclusively, and no acyl transfer was observed, supporting the idea that free copper overrides thiol coordination. We also examined the tyrosine-based system in the presence of an inert thiol, N-Boc-methionine (Supplementary Fig. 18), which lacks nucleophilic reactivity yet can coordinate copper. Notably, the reaction profile remained unchanged compared to the system without the inert thiol (Fig. 2a), suggesting that copper coordination alone is insufficient to alter the reaction sequence without a reactive nucleophile. Overall, the two key factors that govern the observed reaction sequence are: (1) thiol-copper coordination delays catalyst availability, suppressing early CuAAC; and (2) self-assembly, particularly in peptides with aromatic side chains like Ac-CF, slows azide diffusion and prolongs the lifetime of thioester intermediates. Together, these effects allow intrinsic molecular features and supramolecular organization to direct the order of covalent transformations.”

5. In my opinion, the narration would be much easier to understand if the microscopic investigation were presented in parallel with the product distribution of the different systems. The authors may consider revising it accordingly although this is just a suggestion.

Response: We appreciate the reviewer’s suggestion regarding the organization of the microscopy data. The microscopic investigations were intentionally presented alongside the spectroscopic analyses, as both techniques report on changes in supramolecular organization, with fluorescence particularly in the arrangement and evolution of the fluorophore-containing assemblies. This pairing allows a direct correlation between structural features and their spectroscopic signatures. In contrast, the product-distribution profiles reflect the temporal progression of covalent transformations, which does not map always onto nanoscale morphologies. Additionally, the microscopy images, especially for the tyrosine-based peptides show relatively similar aggregate features across systems, whereas the reaction mixtures themselves exhibit clear and significant differences in their temporal evolution and product distributions. These distinctions are thus better conveyed through reaction-progress followed by

discussion on microscopic and spectroscopic data. For these reasons, we believe that keeping the microscopy aligned with the spectroscopic data preserves a clearer narrative flow.

6. SI Fig. 16: The different fluorescence spectra are shown to correspond different species (such as 5a', 6a') but how this was done is not clear from the method section.

Response: We have updated the revised manuscript with a detailed explanation for the same in the method section.

7. SI Fig. 22 & 23: The 72 h TEM images appear to be the same for both cases, although the experiments seem to be different. Please correct this.

Response: We thank the reviewer for pointing this error. This has been corrected in the revised manuscript (Supplementary Fig. 26 and 27).

8. Figure 4: Are there any traces of mono-click product formation?

Response: Indeed, trace amounts (<0.4 mM) of the mono-click product were detected. However, this species rapidly undergoes hydrolysis during the reaction, yielding 2 and preventing significant accumulation. This point is noted in the Supplementary Information beneath Supplementary Fig. 31, and we have ensured it is clearly referenced in the revised manuscript.

9. The conclusion should be emphasized further to better convey the importance of the study.

Response: We appreciate reviewer's suggestion in this regard and the conclusion has been updated accordingly in the revised manuscript with the following text: "We report a one-pot chemical system in aqueous media, in which the sequence of covalent transformations is governed by the structure and reactivity of peptide-based nucleophiles reacting with aminoacyl phosphates bearing alkyne moieties. The phosphate groups act as solubility tags, enabling the dissolution of otherwise hydrophobic alkynes and peptide intermediates, thereby facilitating their participation in aqueous-phase transformations. Depending on the nucleophile, the system selectively prioritizes either acyl transfer or CuAAC, allowing reaction sequence to be directed through molecular design and self-assembly. In tyrosine-containing systems, the low nucleophilicity of phenolates causes CuAAC to outpace acyl transfer, preventing clear temporal separation and yielding mixed triazole-ester outcomes. Thiol-containing substrates, in contrast, rapidly generate thioesters and transiently sequester Cu(I), imposing an ordered sequence in which acyl transfer precedes CuAAC and self-assembly further reinforces this ordering. Incorporating both nucleophiles within a single peptide translates these principles into a three-step cascade, demonstrating that temporal programming can be embedded across multiple covalent transformations. Competition between aliphatic and aromatic azides reveals a consistent preference for aliphatic partners, indicating that azide structure also biases pathway selection. Macroscopic signatures such as gelation and color changes provide real-time readouts of these temporally regulated processes. These transformations proceed without enzymes or external triggers, relying instead on the intrinsic reactivity of nucleophiles, metal sequestration and phase behavior. The reconfiguration of soluble acyl phosphates into esters or thioesters generates dynamic intermediates that modulate both the timing and outcome of subsequent transformations, including the click reaction. The ability to position multiple acyl transfer events upstream of irreversible steps might offer a modular approach to direct reaction cascades and sequences with intrinsic temporal order. Looking ahead, we envision two complementary directions emerging from our study. First, the irreversible click step could, in principle, be replaced or complemented by dynamic reversible chemistries, such as imine formation or thiol-disulfide exchange,

which in turn may introduce adaptability, error correction, and feedback into related reaction networks. Second, the exceptional robustness and orthogonality of CuAAC provide a reliable handle for generating diverse, functionalized aminoacyl phosphate intermediates, which may enable selective and potentially orthogonal acyl-transfer processes even within more complex chemical environments.”